# A Unified Framework for Multimodal Secret Data Steganography

## Abstract

Due to the advances in deep learning and data accessibility, image steganography has become a critical and widely-used tool for information hiding. Image steganography mainly embeds and recovers secret data within cover images. With the increasing variety and volume of data, multimodal secret data steganography is urgently required. However, the framework of existing image steganography often directly embeds multimodal secret information into cover images in a modality-by-modality and sequential manner, leading to unsatisfactory steganography performance. This implies that current image steganography is a modal-specific framework, which is almost effective for hiding the specific modal secret data. **This paper presents a unified framework for multimodal secret data steganography, which is capable of concurrently concealing image, text, and audio data within a cover image and permits reversible recovery.** However, two principal challenges arise: (1) The catastrophic forgetting seriously undermines the consistent performance across various modalities of secret data steganography; (2) The mitigation of catastrophic forgetting further induces significant interference originating from intra- and inter-modal information conflicts among distinct modal secret data and cover images, consequently compromising steganography fidelity. **To achieve coherent multimodal secret data knowledge preservation and interaction, our unified framework firstly establishes a coordinated coupling between steganography tasks and continual learning** to preserve learned multimodal knowledge for maintaining model learning capacity and performance stability. **Subsequently, a Multi-Gap Collaborative Fusion mechanism utilizes cover images as anchors to effectively integrate multimodal knowledge**, resolving intra- and inter-modal conflicts while bolstering security through directed secret data customization and encryption. Experiments demonstrate that our model can achieve secure and high-quality multimodal secret data steganography, outperforming existing state-of-the-art (SOTA) methods.

## 1 Introduction

The widespread adoption of multimodal data across various fields has heightened the need for its secure transmission to prevent unauthorized access. Given the prevalence of image data and rising security requirements, image steganography has emerged as a critical domain within information security. This steganography system is capable of concealing secret data within ordinary cover images with complete visual and statistical imperceptibility, all while guaranteeing lossless data recovery. Conventional image steganography methods, such as Least Significant Bit (LSB) encoding, are mainly designed for text steganography tasks. They usually modify the low-order bits of pixel values to hide secret data.

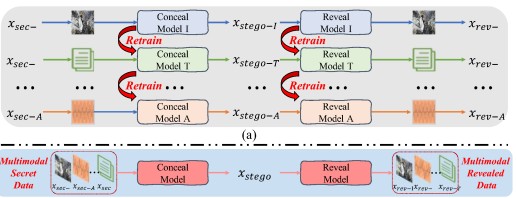

Figure 1: Comparative Analysis: (a) existing models vs. (b) the proposed model. Existing methods are modal-specific and require retraining for each new modal, whereas the proposed model achieves **multimodal secret data concealment within a unified model**.

The advent of Deep Neural Networks (DNNs) has ac-

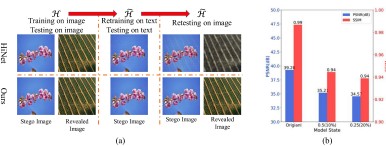

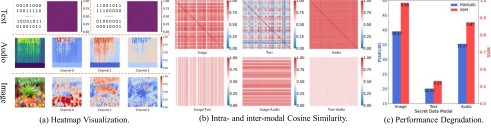

Figure 2: (a) Illustration of Catastrophic Forgetting. The first two columns show performance for image and text data, respectively. Using the text–trained model on image data (right column) shows severe catastrophic forgetting in the baseline, with significant performance drop. Conversely, the proposed method sustains robust cross-modal performance. (b) Under parameter perturbations (±0.5 to 10%; ±0.25 to 20% parameters), steganography performance was significantly compromised, confirming high parameter sensitivity.

Figure 3: Intra- and Inter-modal informational discrepancies, exemplified by text, audio, and image. (a) Heatmap distributions exhibit significant divergences across text, audio, and image modals. (b) Intra-modal and inter-modal cosine similarity exhibit marked dissimilarity patterns across and within modals. (c) The trained model exhibits significant performance degradation on unseen modals. These findings collectively demonstrate underlying informational conflict inherent in multimodal data.

celerated the development of learning-based steganography methods and significantly expanded the variety of concealable data types. Current methods can conceal diverse data, including text Lan et al. (2023); Ma et al. (2025); Xu et al. (2025), image Baluja (2017); Jing et al. (2021); Yu et al. (2024b); Yang et al. (2024); Zhou et al. (2025), audio Soundarya et al. (2018); Krishnan et al. (2025), video Gandikota et al. (2022) and so on. However, current steganography methods are mainly tailored to specific modal and exhibit limited adaptability in increasingly multimodal environments. This limitation forces extensive retraining when encountering new modal data (as depicted in Figure 1), leading to inefficient and non-scalable systems.

To tackle modal-specific constraints and accommodate multimodal environments, **this paper proposes a novel unified multimodal secret data steganography framework that conceals three major modals (image, text, and audio) within cover images using a single model**. However, two critical challenges must be resolved: (1) **Firstly**, multimodal secret data concealment requires the model capable of sustainable learning across diverse modals. Within a steganography model trained on prior modals, data from a novel modality represents a distinct category and constitutes a separate steganography task. When learning a new task (Task $N$), parameter optimization interferes with knowledge acquired from previous tasks (Tasks $1, 2, \cdots, N-1$). However, steganography systems are subject to high parameter sensitivity, *i.e.,* even minor adjustments can disrupt the model's ability to extract previously recoverable data. This interference leads to catastrophic performance degradation on established tasks and complete erosion of system reliability, as illustrated in Figure 2, ultimately resulting in catastrophic forgetting. (2) **Furthermore**, mitigating catastrophic forgetting during concurrent concealing multimodal secret data within a single image induces competition for spatial steganography resources. As illustrated in Figure 3, significant information conflicts exist among these heterogeneous data types. Such conflicts cause substantial intra- and inter-modal interference among the concealed multimodal secret data. This will greatly degrade the steganography fidelity and pose critical security risks to the concealed multimodal secrets. **Consequently, these two issues, coupled with the objective of multimodal secret data steganography, form a self-reinforcing cycle of performance degradation that represents a core challenge in this field.**

To address these issues, the proposed method incorporates **multimodal knowledge preservation and cross-modal interaction**. Inspired by the capacity of continual learning to emulate human lifelong cognitive processes, this study initiates by systematically bridging steganography tasks and continual learning paradigms to overcome single-modal constraints and prevent catastrophic forgetting. This coupling preserves acquired multimodal knowledge, maintains model plasticity, ensures performance stability, and effectively mitigates catastrophic forgetting. Furthermore, both the modal and content of secret data undergo dynamic variation in multimodal secret data steganography. To mitigate both intra- and inter-modal information conflicts within such variable data, a Multi-Gap Collaborative Fusion mechanism is further proposed, which employs cover images, owing to their relative stability, as anchors to directionally refine the secret data. This approach enables targeted customization and encryption of secret data aligned with anchor characteristics, thereby supporting cross-modal interaction while simultaneously reducing conflicts and increasing security. Experimental results demonstrate that the proposed model surpasses SOTA methods in multimodal secret data steganography performance, capacity, and security. The primary contributions of this work are:

- We propose a novel **unified multimodal secret data steganography framework** that firstly achieves simultaneous learning of multimodal secret data steganography tasks within a unified model and delivers superior performance validated by extensive experiments.

- We pioneer achieve **preservation of acquired multimodal knowledge, the sustained retention of learning capacity, and consistency of performance in dynamic multimodal settings** through a structured linkage between image steganography and continual learning.

- We propose a **Multi-Gap Collaborative Fusion** mechanism to directionally refine the multimodal secret data with cover images serve as anchors, thereby enabling cross-modal interaction, mitigating intra- and inter-modal information conflicts, and enhancing security.

## 2 RELATED WORK

### 2.1 IMAGE STEGANOGRAPHY

Image steganography seeks to conceal data within a cover image, ensuring the visual imperceptibility of the stego image and the perfect recovery of the secret. Traditional techniques, such as the LSB Mielikainen (2006) method, were primarily designed for textual data. The advent of Deep Neural Networks (DNNs) has driven the development of learning-based steganography methods, leading to the proposal of high-performance text-hiding methods like SteganoGAN Zhang et al. (2019), FNNS Kishore et al. (2022), LISO Chen et al. (2023), MDDM Xu et al. (2025), and so on. Subsequently, the scope of steganography has expanded beyond text to include image hiding, with techniques such as DDH Baluja (2017), UDH Zhang et al. (2020), various INN-based approaches Lu et al. (2021); Jing et al. (2021); Guan et al. (2022); Zhang et al. (2024a;b); Zhou et al. (2025), and diffusion model-based methods Yu et al. (2024b); Yang et al. (2024). The technology has been expanded to incorporate diverse modals such as audio Soundarya et al. (2018); Huu et al. (2019); Nokhwal et al. (2023); Krishnan et al. (2025), video Gandikota et al. (2022), and so on.

Conventional methods, however, are constrained to static modality configurations, limiting their applicability amidst proliferating multimodal data. In contrast, the proposed framework supports multimodal secret data, thereby significantly broadening its practical applicability.

### 2.2 CONTINUAL LEARNING

Continual learning is a sequential learning framework and aims to empower machine learning models to learn continually from new data, while building upon previously acquired knowledge without forgetting. Formally, given a task sequence $\mathcal{T} = [D^1, D^2, \cdots, D^T]$ of size $T$, where $D^t, 1 \le t \le T$ is the $t$-th task. The dataset for $t$-th task $D_t = \{(x_{t,i}, y_{t,i})\}_{i=1}^{N_t}$ consists of input samples $X_t$ and target samples $Y_t$, where $N_t$ represents the number of samples in the $t$-th task. For a neural network $f$ trained with the task $\mathcal{T}' = [D^1, D^2, \cdots, D^{t-1}]$, the task $D^t$ is a new task. The objective is to learn the new task while maintaining performance on old tasks. Specifically, given an unseen test sample $x \in X$ from any trained tasks, the trained model $f$ should perform well in inferring the label $y = f(x) \in Y$. More releated work is documented in Appendix B.

This work systematically develops a novel framework for continuous multimodal secret data steganography to resolve intra- and inter-modal information conflicts and catastrophic forgetting. Beyond addressing these dual challenges, the integration of Multi-Gap Collaborative Fusion reduces cover-secret discrepancy, thereby mitigating inherent information conflicts.

## 3 PROPOSED METHOD

### 3.1 UNIFIED FRAMEWORK

This work proposes a novel multimodal steganographic framework, as illustrated in Figure 4. Leveraging the established effectiveness of invertible neural networks in frequency-domain steganography Jing et al. (2021); Guan et al. (2022), the proposed method first processes a cover image $x_{cov}$ via discrete wavelet transform (DWT) to derive a latent representation $z_{cov}$. Simultaneously, a concealed payload $x_{sec}$ is transformed into a latent representation $z_{sec}$ of compatible dimensionality.

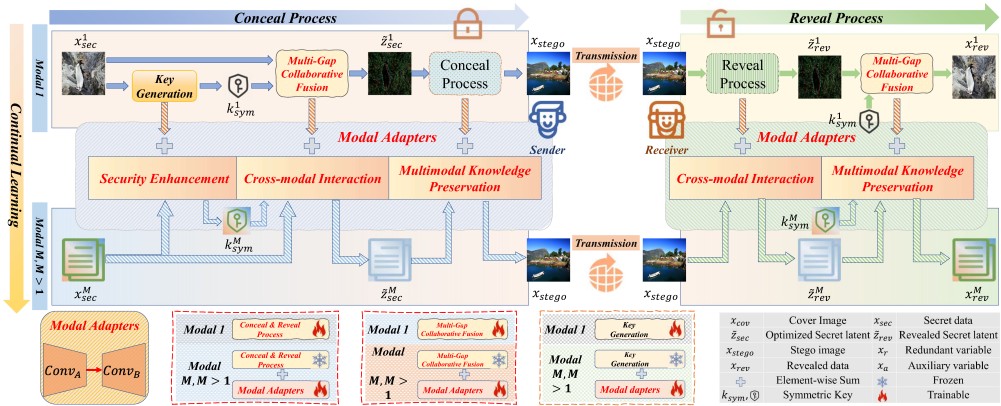

Figure 4: The overall structure of the proposed model. In the conceal stage, multimodal secret data $x_{sec}$ undergoes customization align with anchor $x_{cov}$. The customized secret latent representation $\tilde{z}_{sec}$ is concatenated with the cover image $x_{cov}$ and processed by an invertible neural network (INN)-based steganography network, yielding the stego image $x_{stego}$. The reveal process inversely executes this conceal and optimize pipeline to reveal the multimodal secret data $x_{rev}$.

A Multi-Gap Collaborative Fusion mechanism is subsequently employed to optimize $z_{sec}$ utilizing a symmetric key $k_{sym}$ and the cover latent $z_{cov}$, thereby generating an enhanced latent secret representation $\tilde{z}_{sec}$. This optimized representation $\tilde{z}_{sec}$, along with $z_{cov}$, is processed by the invertible steganography network to produce the stego latent representation $z_{stego}$. The final stego image $x_{stego}$ is reconstructed by applying the inverse wavelet transform (IWT) to $z_{stego}$. The reveal process executes the inverse sequence of operations to recover the original secret payload $x_{rev}$. The entire pipeline is presented in Appendix A.

## 3.2 MULTIMODAL SECRET DATA CONTINUOUS STEGANOGRAPHY

We begin with establishing the definition of task sequence and defining the core problem of multimodal secret data continuous steganography.

*Task Sequence. Let $C = \{x_{cov}^i | 1 \leq i \leq N_t\}$ denote the cover images, $S^t = \{x_{sec}^{t,i} | 1 \leq i \leq N_t\}$ the secret data of the $t$-th modal, and $Y^t = \{x_{stego}^{t,i} | 1 \leq i \leq N_t\}$ the stego images obtained after embedding the secret data of the $t$-th modal, where $N_t$ represents the number of samples in the $t$-th modal. The dataset for the steganography task corresponding to the $t$-th modal is defined as $D^t = \{(x_{cov}^i, x_{sec}^{t,i}, x_{stego}^{t,i} | 1 \leq i \leq N_t\}$, where $t = 1, 2, \cdots, T$. The sequence of multimodal secret data continuous steganography tasks is defined as $\mathcal{T} = [D^1, D^2, \cdots, D^T]$, and the set of task identifiers is given by $\mathbb{T} = [1, 2, \cdots, T]$ and $\forall t \in \mathbb{T}, \mathcal{T}^t = D^t$.*

It should be noted that in cover-based image steganography, the stego image is required to be perceptually indistinguishable from the cover image, and thus could theoretically be represented by the same symbol. However, for clarity and precision in exposition, distinct symbols $C$ and $Y$ are used to denote the set of cover images and the set of stego images, respectively.

In the multimodal environment under investigation, the steganography of secret data from each modal is conceptualized as a distinct task. Consequently, the involved modalities collectively constitute a sequence of steganographic tasks. In the proposed method, image concealing is explicitly designated as the first task within the steganography task sequence.

*Problem Definition. Given a sequence of multimodal secret data to be concealed, consider the secret data associated with the $t$-th modal. Relative to the steganography model trained on data from preceding modals, the data from $t$-th modal constitutes an entirely novel type, thereby defining a new steganography task. The objective of multimodal secret data continuous steganography is to acquire proficiency in this new task while preserving the model's performance on previously learned tasks, thus mitigating catastrophic forgetting. Specifically, for an unseen test sample $x_{sec} \in S$ drawn from any trained task and a cover image $x_{cov} \in C$, the optimized multimodal secret data steganography*

*model $\tilde{\mathcal{H}} : C \times S \rightarrow Y$ should demonstrate effective performance in both the conceal process*

$$x_{stego} = \tilde{\mathcal{H}}(x_{sec}, x_{cov}) \in Y,$$

*and the reveal process*

$$x_{rev} = \tilde{\mathcal{H}}^{-1}(x_{stego}, z_a) \in S,$$

*where $\tilde{\mathcal{H}}^{-1}$ is the reveal process of $\tilde{\mathcal{H}}$.*

This section subsequently describes the multimodal continuous data steganography task into two components: Initial Steganography Task and Forthcoming Steganography Task.

### 3.2.1  INITIAL STEGANOGRAPHY TASK

The initial steganography task, targeting the concealment of a single modal secret data. Following the pipeline of HiNet Jing et al. (2021) and DeepMIH Guan et al. (2022), we utilize the invertible neural network as the base model. For the initial task, unimodal steganography task is implemented via the base model. Cover image spectral coefficients $z_{cov}$ derived via discrete wavelet transform and customized secret data representations $\tilde{z}_{sec}$, constitute the input arguments.

**The conceal process** of the single modal steganography model $\mathcal{H}$ is defined as:

$$\begin{cases} z_{cov}^i & = z_{cov}^{i-1} \odot exp(\alpha(\phi(\tilde{z}_{sec}^{i-1}))) + \psi(\tilde{z}_{sec}^{i-1}), \\ \tilde{z}_{sec}^i & = \tilde{z}_{sec}^{i-1} \odot exp(\alpha(\varphi(z_{cov}^i))) + \chi(z_{cov}^i), \end{cases} \tag{1}$$

where $\odot$ represents the Hadamard product and $exp(\bullet)$ is the exponential function. $z_{cov}$ and $\tilde{z}_{sec}$ denote cover and concealed secrets latent representations, respectively. The scaling factor $\alpha$ implements a sigmoid function scaled by constant $c$. Learnable transformations $\phi(\bullet)$, $\psi(\bullet)$, $\varphi(\bullet)$, and $\chi(\bullet)$ are neural-parameterized functions, instantiated via DenseNet Huang et al. (2017).

The conceal process outputs the stego latent representation $z_{stego}$ and redundant information $z_r$. After that, $z_{stego}$ is transformed back to the spatial domain and obtain the final stego image $x_{stego}$.

**The reveal process** $\mathcal{H}^{-1}$ is the inverse of the conceal process $\mathcal{H}$. The stego image $x_{stego}$ undergoes discrete wavelet transformation to latent representation $z_{stego}$, which is concatenated with a Gaussian noise auxiliary variable $z_a$ as input to the backward reveal process. It is defined as:

$$\begin{cases} \tilde{z}_{sec}^{i-1} & = (\tilde{z}_{sec}^i - \chi(z_{cov}^i)) \odot exp(-\alpha(\varphi(z_{cov}^i))), \\ z_{cov}^{i-1} & = (z_{cov}^i - \psi(\tilde{z}_{sec}^{i-1})) \odot exp(-\alpha(\phi(\tilde{z}_{sec}^{i-1}))). \end{cases} \tag{2}$$

The iterative refinement process described above enables progressive decoupling of concealed secret data from stego latent representation $z_{stego}$.

The reveal and conceal processes utilize identical architectural configurations and parameters. The reveal process outputs customized and encrypted payloads $\tilde{z}_{rev}$, which undergo subsequent decryption to yield the final revealed payloads $x_{rev}$, with the decryption process detailed later.

### 3.2.2  FORTHCOMING STEGANOGRAPHY TASK

Upon completion of the initial steganography task, training proceeds sequentially through the remaining tasks. Following this procedure, each forthcoming steganography task incorporates the complete parameter set from the initial steganography model $\mathcal{H}$ to obtain the final model $\tilde{\mathcal{H}}$.

Spcifically, to facilitate multimodal secret data steganography, modal adaptivers are incorporated into the original unimodal steganography model. These layers enable the acquisition of novel modal information while preserving capabilities learned from the original modality. The modal adaptivers are implemented using LoRA Hu et al. (2022): specifically, a lightweight LoRA convolution layer is appended to each standard convolution layer to capture new modal features. The output of the modal adaptive convolution layer is given by:

$$\mathcal{O}_{cov}(z) = Conv(z) + \beta \cdot LoRA(z), \tag{3}$$

where $\beta \in \mathbf{R}^{1 \times C \times 1 \times 1}$ is the learnable scaling parameter and $z$ is the input of the convolution layer corresponds to either the cover image or the secret data.

The LoRA layer is implemented via two convolutional layers, as depicted in Hu et al. (2022). The output of this layer is expressed as follows:

$$LoRA(z) = Conv(SiLU(Conv(z))).$$ (4)

During training, solely the LoRA convolution parameters and scaling parameter $\beta$ are optimized, while the parameters retained from the initial steganography model remain invariant.

The remaining settings of $\tilde{\mathcal{H}}$ and $\tilde{\mathcal{H}}^{-1}$ are preserved from the base initial steganography model $\mathcal{H}$, enabling multimodal secret data steganography with negligible structural alterations.

### 3.3 Multi-Gap Collaborative Fusion

Within the described multimodal secret data continuous steganography pipeline, the secret data from distinct modals exhibit significant intra- and inter-modal information conflicts. These conflicts induce substantial information interference, which severely compromises the quality of steganography images and the accuracy of secret data extraction. Consequently, mitigating these conflicts is essential for high-fidelity steganography.

Given that the modalities and content of the secret data undergo continuous variation, the resulting information conflicts are inherently dynamic. Thus, addressing these conflicts requires a stable anchor point. The cover image $x_{cov}$, being relatively fixed, serves as this natural anchor. By directionally customizing and optimizing the multimodal secret data align with this anchor to minimize their disparity, the information conflicts can be effectively mitigated. Based on the cover image $x_{cov}$, we propose a Multi-Gap Collaborative Fusion mechanism $F$ to directionally customize the secret data $x_{sec}$. As shown in Figure 5, the forward customization process is defined as

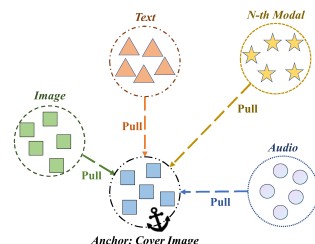

Figure 5: Illustration of Multi-Gap Collaborative Fusion.

$$\tilde{z}_{sec} = F(x_{sec}, x_{cov}, k_{sym}), \; s.t \; d(\tilde{z}_{sec}, x_{cov}) \leq d(x_{sec}, x_{cov}),$$ (5)

where $k_{sym}$ is the symmetric key generated with the cover image $x_{cov}$ and concealed payloads $x_{sec}$ following the pipeline in Wang et al. (2025). Distance function $d(\bullet)$ serves to compute the distance between $x_{sec}$ and $\tilde{z}_{sec}$ against the anchor $x_{cov}$. While retaining the original pipeline, the AlexNet Krizhevsky et al. (2012) is replaced with SHA-256 hashing to calculate the hash value of secret data and cover image, respectively. This modification enables multimodal payload classification while reducing computational complexity. To recover the secret information $x_{rev}$, the output $\tilde{z}_{rev}$ generated by the reveal process $\tilde{\mathcal{H}}^{-1}$ is leveraged within the backward restoration process $F^{-1}$ of the forward customization process follows:

$$x_{rev} = F^{-1}(\tilde{z}_{rev}, x_{cov}, k_{sym}).$$ (6)

This mechanism minimizes the divergence between the cover image and the secret data, enhancing the compatibility of the secret data for concealing within the cover. Since multi-modal secret data are optimized relative to the same anchor point, the customized data converge toward that point, thereby reducing mutual information conflicts among them.

Specifically, the Multi-Gap Collaborative Fusion mechanism adopts an invertible architecture consistent with that of the steganography network. Crucially, while the steganography network generates cover image as forward output, this mechanism yields customized secret data $\tilde{z}_{sec}$. Meanwhile, a symmetric key (detailed in the Appendix C) is incorporated within this mechanism to enhance secret data security. This enables simultaneous secret data customization and encryption.

**The forward customization process** $F$ is formulated as:

$$\begin{cases} z_{cov}^i = z_{cov}^{i-1} \odot exp(\alpha(\eta(k_{sym}) \odot \phi(\tilde{z}_{sec}^{i-1}))) + \psi(\tilde{z}_{sec}^{i-1}), \\ \tilde{z}_{sec}^i = \tilde{z}_{sec}^{i-1} \odot exp(\alpha(\eta(k_{sym}) \odot \varphi(z_{cov}^i))) + \chi(z_{cov}^i), \end{cases}$$ (7)

and $\tilde{z}_{sec}^0 = z_{sec}$, the non-optimized concealed payloads. The key-related weight generation module $\eta(\bullet)$ with symmetric key $k_{sym}$ as input, is defined as:

$$\eta(k_{sym}) = Conv(\theta) \odot Conv(SiLU(Conv(k_{sym}))), \; \theta \in \mathbf{R}^{B \times C \times H \times W}.$$ (8)

The customized secret data $\tilde{z}_{sec}$ and cover image latent representation $z_{cov}$ are subsequently processed by the steganography network $\tilde{\mathcal{H}}$ to synthesize the latent $z_{stego}$ of the stego image $x_{stego}$.

**The backward restoration process** $F^{-1}$, which constitutes the inverse operation of the previously described forward customization process, is:

$$\begin{cases} \tilde{z}_{sec}^{i-1} = (\tilde{z}_{sec}^{i} - \chi(z_{cov}^{i})) \odot exp(-\alpha(\eta(k_{sym}) \odot \varphi(z_{cov}^{i}))), \\ z_{cov}^{i-1} = (z_{cov}^{i} - \psi(\tilde{z}_{sec}^{i-1})) \odot exp(-\alpha(\eta(k_{sym}) \odot \phi(\tilde{z}_{sec}^{i-1}))). \end{cases} \quad (9)$$

During the initial phase of the restoration process, auxiliary variable $z_{aux}$ must be introduced to maintain dimensional compatibility within the invertible neural network. However, discrepancies between these auxiliary variable $z_{aux}$ and the redundant variable $z_r$ ultimately generated during the forward encryption process can degrade the quality of the decrypted secret information.

To mitigate this issue, the frequency representation of the cover image $z_{cov}$ is utilized as the introduced auxiliary variable. Furthermore, to ensure consistency between the redundant variable $z_r$ and auxiliary variable $z_{aux}$, the constraint is imposed on the redundant variable produced in the forward customization process. This constraint, denoted as *Encryption loss $\mathcal{L}_e$*, will be detailed later.

### 3.4 Loss Function

Our loss function comprises conceal loss, reveal loss, encryption loss, low-frequency wavelet loss:

**Conceal loss.** The steganography process outputs the stego image $x_{stego}$ based on the cover image $x_{cov}$ and secret image $x_{sec}$. For security purposes, the stego image $x_{stego}$ should closely match the cover image $x_{cov}$ making them indistinguishable. Thus, we define the conceal loss as:

$$\mathcal{L}_c = l_s(x_{stego}, x_{cov}). \quad (10)$$

Besides, the low-frequency wavelet loss $\mathcal{L}_f$ utilized to conceal the secret data within the high-frequency region of the cover image is formulated as:

$$\mathcal{L}_f = l_s(D_{LL}(x_{stego}), D_{LL}(x_{cov})), \quad (11)$$

where $l_s$ represents the $l_1$ or $l_2$ norm, serving as a measure of the difference between two latents. In our experiments, we use the $l_2$ norm as the default.

**Reveal loss.** To ensure that the revealed data $x_{rev}$ aligns with the secrets $x_{sec}$, the reveal loss is:

$$\mathcal{L}_r = l_s(x_{rev}, x_{sec}). \quad (12)$$

**Encryption loss.** The encryption loss function $\mathcal{L}_e$ minimizes the reconstruction error between decrypted secret data $x_{rev}$ and original secret data $x_{sec}$, where error originates from the auxiliary variable $z_{aux}$ introduction during decrypt the secret data. It is formulated as:

$$\mathcal{L}_e = l_s(z_{cov}, z_{aux}). \quad (13)$$

**Total loss.** The total loss function $\mathcal{L}_{Total}$ is the weighted sum of the conceal loss $\mathcal{L}_c$, reveal loss $\mathcal{L}_r$, encryption loss $\mathcal{L}_e$ and low-frequency wavelet loss $\mathcal{L}_f$, formulated as:

$$\mathcal{L}_{Total} = \lambda_1 \mathcal{L}_c + \lambda_2 \mathcal{L}_r + \lambda_3 \mathcal{L}_e + \lambda_4 \mathcal{L}_f, \quad (14)$$

where $\lambda_1$, $\lambda_2$, $\lambda_3$ and $\lambda_4$ are trade-off parameters set to 2.0, 1.0, 0.5 and 1.0, respectively, for balance.

## 4 Experiments

### 4.1 Experimental Setting

Our model is implemented with PyTorch and trained on the DIV2K Agustsson & Timofte (2017) training dataset. The evaluation is performed on the DIV2K Agustsson & Timofte (2017) test dataset, COCO Lin et al. (2014), and ImageNet Russakovsky et al. (2015) at a resolution of $256 \times 256$. More implementation details are presented in the Appendix D.

Table 1: Numerical comparisons with different steganography methods across various datasets, highlighting the best results in **bold** and the second-best in underline.

| Method | Time(s) | Cover/Stego | | | | | | | | | | | |
| | | Image(DIV2K) | | | | Text(3bpp) | | | | Audio | | | |
| | | PSNR↑ | SSIM↑ | MAE↓ | RMSE↓ | PSNR↑ | SSIM↑ | MAE↓ | RMSE↓ | PSNR↑ | SSIM↑ | MAE↓ | RMSE↓ |
|---|---|---|---|---|---|---|---|---|---|---|---|---|---|
| SteganoGAN | 0.04 | - | - | - | - | 21.22 | 0.6124 | 16.59 | 22.53 | - | - | - | - |
| FNNS-D | 5.95 | - | - | - | - | 23.02 | 0.6907 | 13.82 | 18.47 | - | - | - | - |
| LISO | 0.08 | - | - | - | - | 30.44 | 0.8541 | 5.85 | 7.88 | - | - | - | - |
| VoI-GAN | 2.86 | - | - | - | - | - | - | - | - | 34.86 | 0.8457 | 4.86 | 7.89 |
| ASA | 1.89 | - | - | - | - | - | - | - | - | 42.54 | 0.9858 | 1.94 | 2.87 |
| ISN | 0.23 | 39.28 | 0.9853 | 2.34 | 2.91 | 19.45 | 0.5403 | 21.87 | 27.75 | 36.31 | 0.9585 | 2.81 | 3.93 |
| HiNet | 0.18 | 39.53 | 0.9868 | 2.08 | 2.87 | 20.10 | 0.5372 | 19.71 | 26.40 | 37.08 | 0.9575 | 2.73 | 3.65 |
| DeepMIH | 0.15 | 43.72 | 0.9895 | 1.94 | 2.81 | 20.91 | 0.5861 | 17.91 | 24.40 | 39.15 | 0.9681 | 2.54 | 3.57 |
| iSCMIS | 0.17 | 45.78 | 0.9924 | 1.62 | 2.42 | 21.14 | 0.5927 | 17.44 | 23.91 | 40.61 | 0.9714 | 2.46 | 3.49 |
| StegFormer | 0.15 | 48.08 | 0.9963 | 1.51 | 2.37 | 21.20 | 0.5989 | 17.31 | 23.74 | 41.98 | 0.9725 | 2.22 | 3.13 |
| Ours | 0.16 | **50.72** (2.64↑) | **0.9987** (0.0024↑) | **0.55** (0.96↓) | **0.78** (1.59↓) | **42.35** (11.91↑) | **0.9951** (0.1451↑) | **1.46** (4.39↓) | **1.99** (5.89↓) | **45.51** (2.97↑) | **0.9965** (0.0107↑) | **1.01** (0.93↓) | **1.39** (1.48↓) |

| Method | Time(s) | Secret/Reveal | | | | | | | | |
| | | Image(DIV2K) | | | | Text(3bpp) | Audio | | | |
| | | PSNR↑ | SSIM↑ | MAE↓ | RMSE↓ | Error Rate (%) ↓ | PSNR↑ | SSIM↑ | MAE↓ | RMSE↓ |
|---|---|---|---|---|---|---|---|---|---|---|
| SteganoGAN | 0.03 | - | - | - | - | 13.74 | - | - | - | - |
| FNNS-D | 4.62 | - | - | - | - | 0.10 | - | - | - | - |
| LISO | 0.07 | - | - | - | - | 2E-03 | - | - | - | - |
| VoI-GAN | 2.37 | - | - | - | - | - | 45.37 | 0.9698 | 2.83 | 3.53 |
| ASA | 1.30 | - | - | - | - | - | 45.02 | 0.9651 | 2.97 | 3.79 |
| ISN | 0.23 | 37.06 | 0.9672 | 2.80 | 4.30 | 19.45 | 35.63 | 0.9567 | 3.49 | 5.87 |
| HiNet | 0.18 | 46.64 | 0.9962 | 0.93 | 1.31 | 20.87 | 38.07 | 0.9696 | 2.67 | 4.04 |
| DeepMIH | 0.15 | 42.56 | 0.9851 | 1.94 | 2.91 | 20.58 | 36.30 | 0.9501 | 3.27 | 5.41 |
| iSCMIS | 0.17 | 42.53 | 0.9836 | 2.11 | 3.04 | 17.39 | 37.07 | 0.9608 | 2.83 | 4.30 |
| StegFormer | 0.15 | 48.25 | 0.9961 | 1.47 | 2.38 | 18.11 | 40.01 | 0.9617 | 2.59 | 3.92 |
| Ours | 0.16 | **53.10** (4.85↑) | **0.9996** (0.0034↑) | **0.41** (0.52↓) | **0.61** (0.7↓) | **0** (2E-03↓) | **46.58** (1.21↑) | **0.9947** (0.0249↑) | **0.89** (1.7↓) | **1.00** (2.53↓) |

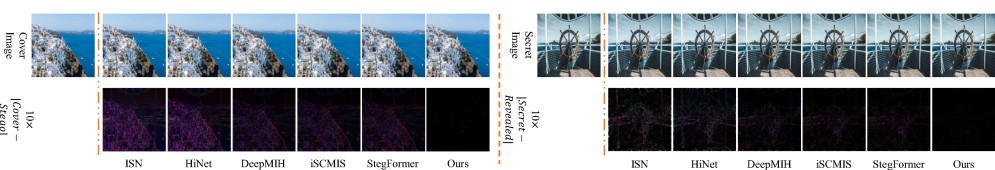

Figure 6: Visual comparisons of stego images and revealed secret images for the proposed model and various steganography models on the DIV2K dataset.

## 4.2 QUALITY ANALYSIS

**Quantitative results**. The steganography performance was initially evaluated on DIV2K datasets, with comprehensive results detailed in Table 1. On the DIV2K benchmark, our model demonstrates significant improvements: PSNR increases by **2.64dB** and SSIM by **0.24%** for cover/stego image pairs, while PSNR rises by **4.85dB** and SSIM by **0.34%** for secret/revealed secret pairs. Concurrently, MAE and RMSE exhibit reductions of **0.96/1.59** and **0.52/0.7** for these respective pairs. Furthermore, the table presents the time consumed of various methods, and the proposed method demonstrates comparable time efficiency. **More detailed efficiency analysis is presented in Appendix E.** These results illustrate that the proposed model is associated with significant improvements in the quality of both the stego and the revealed secret images relative to other methods.

The proposed method was also evaluated on *text and audio datasets*. As demonstrated in Table 1, for text secret data, the proposed method yields an improvement of **11.91dB** in PSNR and a **14.51%** gain in SSIM for cover/stego image pairs. The extracted secret data also exhibits a reduced error rate. In the case of audio data, the method demonstrates enhanced steganographic performance and achieves a **0.53dB** increase in PSNR and a **0.4%** increase in SSIM for cover/stego pairs. Furthermore, for extracted audio spectrograms, it delivers superior results, with PSNR and SSIM gains of **1.21dB** and **0.3%** for secret/reveal pairs, respectively. Concurrently, it reduces both MAE and RMSE for all cover/stego and secret/reveal data pairs. These results indicate that the proposed method achieves robust steganography performance across diverse data modals, highlighting its strong capability and adaptability in multimodal environments.

**Qualitative Results.** The Figure 6 assesses the visual results of image-in-image steganography and presents the stego and recovered images generated by various methods. The figure also displays residual maps between the cover/stego and secret/revealed image pairs. The results demonstrate that the proposed method produces the smallest residuals, confirming its superiority in generating

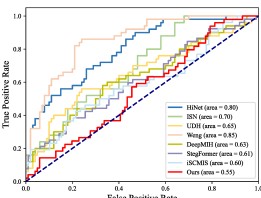

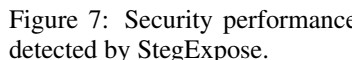

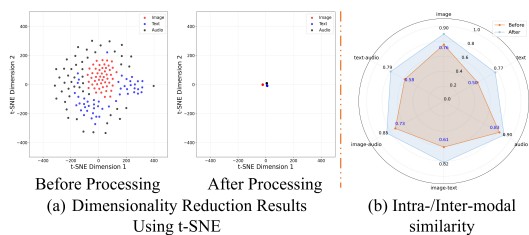

Before Processing     After Processing
(a) Dimensionality Reduction Results Using t-SNE     (b) Intra-/Inter-modal similarity

Figure 7: Security performance detected by StegExpose.

Figure 8: Comparison of information conflicts before/after MGCF processing.

Table 2: The detection accuracy (%) detected by SRNet, XuNet and YeNet.

| | Weng | UDH | ISN | HiNet | Deep MIH | iSC MIS | Steg Former | **Ours** |
|---|---|---|---|---|---|---|---|---|
| SRNet | 89.25 | 85.31 | 84.91 | 79.32 | 75.54 | 69.64 | 58.39 | **53.98(4.41↓)** |
| XuNet | 82.24 | 79.26 | 77.42 | 75.37 | 74.12 | 67.86 | 57.23 | **55.04(2.19↓)** |
| YeNet | 85.18 | 82.13 | 80.27 | 77.86 | 69.24 | 68.92 | 58.03 | **54.61(3.42↓)** |

Table 3: Effectiveness of Secret optimize module and Encryption Loss $\mathcal{L}_e$.

| MGCF | $\mathcal{L}_e$ | Cover/Stego | | | | Srcret/Reveal | | | |
|---|---|---|---|---|---|---|---|---|---|
| | | PSNR | SSIM | MAE | RMSE | PSNR | SSIM | MAE | RMSE |
| ✗ | ✗ | 43.72 | 0.9895 | 1.94 | 2.81 | 42.56 | 0.9851 | 1.94 | 2.91 |
| ✓ | ✗ | 45.16 | 0.9969 | 1.08 | 1.54 | 48.41 | 0.9975 | 0.76 | 1.09 |
| ✓ | ✓ | **50.72** | **0.9987** | **0.55** | **0.78** | **53.10** | **0.9996** | **0.41** | **0.61** |

higher-quality stego images and achieving more accurate secret image reconstruction. **More results are presented in the Appendix F and Appendix H.** These results exemplify that the proposed model achieves notable improvements in effectiveness and security over existing SOTA models.

### 4.3 ABLATION STUDIES

**Steganographic analysis.** To evaluate the anti-steganalysis capability of various methods, we employ StegExpose Boehm (2014) and three steganalysis networks: SRNet Boroumand et al. (2018), XuNet Xu et al. (2016), and YeNet Ye et al. (2017). Lower detection accuracy and a smaller area under curve (AUC) indicates better security performance. The evaluate results are presented in Figure 7 and Table 2 respectively. These steganalysis results indicate that the proposed model achieves superior anti-steganalysis performance compared to other SOTA methods.

**Effect of the Multi-Gap Collaborative Fusion (MGCF).** As illustrated in Table 3, the introduced MGCF mechanism significantly enhances steganography performance. Quantitative analysis demonstrates PSNR improvements of **1.44dB** for cover/stego image pairs and **5.85dB** for secret/revealed image pairs, with corresponding SSIM gains of **0.74%** and **1.24%** respectively. Concurrently, the module reduces MAE and RMSE metrics for both image pairs. These highlight the crucial role of the introduced Multi-Gap Collaborative Fusion mechanism in strengthening the steganography performance. **Further analysis are provided in Appendix G.**

**Mitigate Information Conflicts.** The MGCF is proposed to mitigate the intra- and inter-modal information conflict in multimodal secret data. It leverages the cover image as an anchor to achieve targeted customization of the secret data. To validate its efficacy, relevant experiments were conducted with a fixed cover image and 100 samples per modality. Figure 8 presents the t-SNE visualization and cosine similarity measurements of data samples before and after MGCF processing. The results confirm that the proposed directed customization significantly reduces feature divergence and enhances similarity across the secret data. This reduction in feature divergence, coupled with a significant improvement in steganography performance, demonstrates the mechanism's effectiveness in alleviating both intra- and inter-modal information conflicts.

## 5 CONCLUSION

This paper proposes a novel multimodal secret data steganography framework that enables concurrent concealment and recovery of multimodal secret data within a unified architecture. It achieves this outcome through multimodal knowledge preservation and cross-modal interaction. By establishing a systematic coupling between steganography tasks and continuous learning, the method effectively retains acquired multimodal knowledge and sustains learning capability. Furthermore, leveraging the cover image as an anchor, the proposed method performs targeted customization of secret data through Multi-Gap Collaborative Fusion mechanism. This process concurrently enables cross-modal interaction and mitigates inherent intra- and inter-modal information conflicts among the multimodal secret data. Empirical evaluations demonstrate the model's superior performance over existing steganography models in multimodal secret data steganography tasks.

## 6 REPRODUCIBILITY STATEMENT

During the publication phase, we will provide full access to all codes, logs, and result files to ensure transparency and reproducibility of our work.

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

## A    PIPELINE

---

**Algorithm 1** The Concealing Process

---

**Require:** The secret image $x_{sec}$ which will be concealed, the cover image $x_{cov}$, trained Multi-Gap Collaborative Fusion $F$ and multimodal secret data steganography model $\tilde{\mathcal{H}}$, Discrete Wavelet Transform (DWT), Inverse Wavelet Transform (IWT), SHA-256 hash function, MLP for private key generation, the weight $W$ for public key derivation.

**Ensure:**

$z_{sec} = DWT(x_{sec}), z_{cov} = DWT(x_{cov})$

$H_s \leftarrow SHA - 256(x_{sec}), H_c \leftarrow SHA - 256(x_{cov})$
$k_{pri-i} \leftarrow MLP(H_i), i \in s, c$
$k_{pub-s} \leftarrow W \cdot k_{pri-s}$
$k_{sym} \leftarrow k_{pub-s} \cdot k_{pri-c}$    # Symmetric key generation.
$\tilde{z}_{sec} = F(x_{sec}, x_{cov}, k_{sym})$    # Directionally customize the secret data.
$z_{stego} = \tilde{\mathcal{H}}(z_{cov}, \tilde{z}_{sec})$    # Conceal the customized secret data with steganography model.
$x_{stego} \leftarrow IWT(z_{stego})$

---

**Algorithm 2** The Revealing Process

---

**Require:** The stego image $x_{stego}$ which contains the secret data, the cover image $x_{cov}$, trained Multi-Gap Collaborative Fusion $F$ and multimodal secret data steganography model $\tilde{\mathcal{H}}$, Discrete Wavelet Transform (DWT), Inverse Wavelet Transform (IWT), public key of secret image $k_{pub-s}$, random Gaussian noise $z_{aux}$, SHA-256 hash function, MLP for private key generation, the weight $W$ for public key derivation.

**Ensure:**

$z_{stego} = DWT(x_{stego})$
$\tilde{z}_{rev} = \tilde{\mathcal{H}}^{-1}(z_{stego}, z_{aux})$    # Reveal the customized secret data with steganography model.
$H_c \leftarrow SHA - 256(x_{cov})$
$k_{pri-c} \leftarrow MLP(H_c)$
$k_{pub-s} \leftarrow W \cdot k_{pri-s}$
$k_{sym} \leftarrow k_{pub-s} \cdot k_{pri-c}$    # Symmetric key generation.
$z_{rev} = F^{-1}(\tilde{z}_{rev}, x_{cov}, k_{sym})$    # Directionally customize the secret data.
$x_{sec} = IWT(z_{rev})$

---

## B    RELATED WORK

### B.1    IMAGE STEGANOGRAPHY

Traditional image steganography methods, such as Least Significant Bits (LSB) Mielikainen (2006), Pixel Value Differencing (PVD) Pan et al. (2011), and so on, accomplish the concealment of secret information by embedding it into the pixel space of an image or a certain transform space (e.g., Discrete Fourier Transform (DFT), and Discrete Wavelet Transform (DWT)).

The advancement of deep learning has spurred growing interest in deep learning-based image steganography methods, which have exhibited superior performance in steganography tasks. These methods substantially improve embedding capacity and broaden the range of concealed data types.

In the domain of text hiding, SteganoGAN Zhang et al. (2019) employs a generative adversarial network (GAN) framework and utilizes adversarial training to accomplish steganography objectives. FNNS Kishore et al. (2022) capitalizes on the sensitivity of neural networks to subtle perturbations, generating adversarially perturbed images to guarantee precise information recovery. LISO Chen et al. (2023) introduces a novel gradient-based neural optimization algorithm that integrates the capacity of neural networks to learn image manifolds with the precision of constrained optimization. MDDM Xu et al. (2025) encodes the secret message into the initial noise for image generation by

utilizing a Cardan grille and leverages the reversibility of DDIM to develop a message-driven image steganography framework based on diffusion models.

The increasing prevalence of diverse data types in social media has created a growing need for steganography techniques capable of handling multiple modalities. For image, Baluja was the first to achieve the steganography of an entire secret image through the encoder-decoder neural network DDH Baluja (2017; 2019). UDH Zhang et al. (2020), on the other hand, provided a different perspective from DDH for the image steganography task. Subsequently, with the development of reversible neural networks and flow models, researchers turned to using reversible neural networks to achieve high-capacity image steganography, and ISN Lu et al. (2021) was the earliest work in this regard. In subsequent research Jing et al. (2021); Guan et al. (2022); Zhang et al. (2024a;b); Zhou et al. (2025), relevant researchers explored multi-image steganography using Invertible Neural Networks, increasing the steganographic capacity once again. CRoSS Yu et al. (2024b) and Diff-Stega Yang et al. (2024) use diffusion models to achieve the steganography of secret images. They use text prompts and image prompts to guide the generation of stego images, enabling stable and controllable generative steganography.

Steganography research has extended beyond text and image data to encompass various other modalities, including Soundarya et al. (2018); Huu et al. (2019); Nokhwal et al. (2023); Krishnan et al. (2025), video Gandikota et al. (2022), watermark Lukas et al. (2024); Garov et al. (2024); Arabi et al. (2025) and so on, with studies across these domains demonstrating effective performance.

Conventional methods, however, are constrained to static modality configurations, limiting their applicability amidst proliferating multimodal data. In contrast, the proposed framework supports multimodal secret data, thereby significantly broadening its practical applicability.

## B.2 CONTINUAL LEARNING

Continual learning is designed to equip models with the capacity for sequential knowledge acquisition when encountering new data distributions or tasks, while avoiding catastrophic forgetting of previously acquired knowledge. Its fundamental objective lies in achieving a balance between model stability—the retention of prior knowledge—and plasticity, the integration of new knowledge, thereby approximating human lifelong cognitive processes.

Formally, given a task sequence $\mathcal{T} = [D^1, D^2, \cdots, D^T]$ of size $T$, where $D^t, 1 \leq t \leq T$ is the $t$-th task. The dataset for $t$-th task $D_t = \{(x_{t,i}, y_{t,i})\}_{i=1}^{N_t}$ consists of input samples $X_t$ and target samples $Y_t$, where $N_t$ represents the number of samples in the $t$-th task. For a neural network $f$ trained with the task $\mathcal{T}' = [D^1, D^2, \cdots, D^{t-1}]$, the task $D^t$ is a new task. The objective is to learn the new task while maintaining performance on old tasks. Specifically, given an unseen test sample $x \in X$ from any trained tasks, the trained model $f$ should perform well in inferring the label $y = f(x) \in Y$.

In previous work, Shi et al. (2025) propose a dual-representation mechanism that emulates specific and generalized memory systems, substantially mitigating catastrophic forgetting while reducing computational energy consumption. Lu et al. Lu et al. (2025) establish a stability–plasticity equilibrium through synergistic interactions between deep-narrow architectures, optimized for plasticity, and wide-shallow structures, enhanced for stability. The framework D-MoLE Ge et al. (2025) employs gradient-guided dynamic parameter allocation to enable on-demand resource adaptation across visual and textual modalities. Additionally, Wang et al. (2024) provide a comprehensive survey of continual learning, aiming to establish connections between fundamental settings, theoretical foundations, representative methods, and practical applications. Yu et al. (2024a) begin by outlining essential background knowledge in multimodal continual learning and proceed to conduct the first systematic review dedicated to this emerging field. Pan et al. (2025) provides a comprehensive review of contemporary applications of continual learning within reinforcement learning.

This work systematically develops a novel framework for continuous multimodal secret data steganography to resolve inter-modal information conflicts and catastrophic forgetting. Beyond addressing these dual challenges, the integration of cover-secret gap reduction and encryption reduces cover-secret discrepancy, thereby mitigating inherent information conflicts.

## C  KEY GENERATION

Building upon Wang et al. (2025), symmetric key $k_{sym}$ is derived with cover image and concealed payloads, which is formulated as:

$$
\begin{aligned}
k_{sym} &= k_{pub-s} \cdot k_{pri-c} \\
&= ((W_L + W_S) \cdot k_{pri-s}) \cdot k_{pri-c} \\
&= ((W_L + W_S) \cdot k_{pri-c}) \cdot k_{pri-s} \\
&= k_{pub-r} \cdot k_{pri-s} \,,
\end{aligned}
\tag{15}
$$

where $W_L$ denote the MLP weight matrix employed for private key generation, and $W_S$ is the weight associated with secret data. The private keys are derived as $k_{pri-i} = MLP(F_i)$ for $i \in c, s$, yielding $k_{pri-c}$ and $k_{pri-s}$ as private keys for cover image $x_{cov}$ and secret data $x_{sec}$ respectively.

While retaining the original pipeline, the AlexNet Krizhevsky et al. (2012) —previously extracting $F_s$ (payload features) and $F_c$ (cover features)—is replaced with SHA-256 hashing to calculate the hash value $H_s$ and $H_c$ of secret data and cover image, respectively. This modification enables multimodal payload classification while reducing computational complexity.

During stego image ($x_{stego}$) transmission, the associated public key ($k_{pub-s} = (W_L + W_S) \cdot k_{pri-s}$) is transmitted concurrently. The receiver employs the pipeline defined in Equation (15) to derive the symmetric key ($k_{sym}$), identical to the sender's concealment procedure. This facilitates accurate secret data revealing, completing the reveal process. Following the previous work Wang et al. (2025), we generate the symmetric key $k_{sym}$ with the cover image and the secret data. The pipeline is the same as the method introduced in Wang et al. (2025) and we replace the AlexNet utilized to extract the features $F_s$ and $F_c$ from the secret and cover image, respectively, with the SHA-256 hash function. This can not only classify the secret data, but also satisfy the multimodal secret data. Besides, it also can reduce the computation flops.

The aforementioned key generation mechanism is constructed following the ECDHE algorithm. A comparative overview of the standard ECDHE process and the implemented protocol is presented below:

**(1) The Principle of the ECDHE:**

If we have two private keys $k_{pri-a}$ and $k_{pri-b}$ (belonging to A and B, respectively) and an ECC elliptic curve with generator point $G$, we can exchange over an insecure channel the values $k_{pub-a} = k_{pri-a} \cdot G$ and $k_{pub-b} = k_{pri-b} \cdot G$ (the public keys of A and B) and then we can derive a shared secret symmetric key: $k_{sym} = k_{pub-b} \cdot k_{pri-a} = k_{pub-a} \cdot k_{pri-b}$. The ECDH algorithm is trivial:

- A generates a random ECC key pair: $k_{pri-a}, k_{pub-a} = k_{pri-a} \cdot G$.
- B generates a random ECC key pair: $k_{pri-b}, k_{pub-b} = k_{pri-b} \cdot G$.
- A and B exchange their public keys through the insecure channel (e.g. over Internet).
- A calculates $k_{sym} = k_{pub-b} \cdot k_{pri-a}$.
- B calculates $k_{sym} = k_{pub-a} \cdot k_{pri-b}$.

Now both A and B have the same key $k_{sym} = k_{pub-b} \cdot k_{pri-a} = k_{pub-a} \cdot k_{pri-b}$.

**(2) Our implementation:**

- The sender (A) and receiver (B) share the same cover image for image concealment.
- A calculates the private keys $k_{pri-i} = MLP(H_i), i \in s, c$, the public key related to the secret image $k_{pub-s} = W \cdot k_{pri-s}$ and the symmetric key $k_{sym} = k_{pub-s} \cdot k_{pri-c}$. Here, $W = W_L + W_S$, where $W_L$ represents static parameters derived from a fixed linear layer, and $W_S$ denotes dynamic parameters generated based on the secret image. This process is analogous to the selection of specific base points ($G$) and elliptic curves ($E$) within the ECDHE algorithm.
- B calculates the private keys $k_{pri-c} = MLP(H_c)$.
- A transmits the public key $k_{pub-s}$ to B.

- B calculates the symmetric key $k_{sym} = k_{pub-s} \cdot k_{pri-c}$.
- B reveals the secret image with $k_{sym}$.

Throughout the steganography process, only the stego image and the public key associated with the secret image are transmitted, with all the private key remaining undisclosed. Consequently, the proposed key generation mechanism maintains compliance with the ECDHE protocol and retains its provable security guarantees.

## D  IMPLEMENTATION DETAILS

**Datastes and Setting.** The model is implemented in PyTorch and trained on the DIV2K Agustsson & Timofte (2017) training dataset. The evaluation is performed on the DIV2K Agustsson & Timofte (2017) test dataset(100 images), COCO Lin et al. (2014) (5000 images), and ImageNet Russakovsky et al. (2015) (10,000 images). Training images are randomly cropped to $256 \times 256$ and augmented with random horizontal and vertical flips. Comparatively, test images in the DIV2K dataset are center-cropped, while in the other datasets, the images are resized to $256 \times 256$. The AdamW optimizer with an initial learning rate of $1 \times 10^{-5}$ is used for training.

The text data comprises randomly-generated binary data. For the steganography capacity of 3 bits per pixel (3 bpp) examined in this study, the data consists of 196,608 characters. This data is subsequently reconfigured into a three-dimensional array of size $3 \times 256 \times 256$, which serves as the input to the subsequent steganography process.

The audio data was obtained from the publicly accessible Dani-Voice dataset. Following the standard methodology for audio information hiding, the data was converted into spectrogram representations via the Short-Time Fourier Transform (STFT) to facilitate subsequent embedding and extraction operations. The steganography performance was subsequently assessed based on these spectrograms. All experiments are conducted on a Nvidia 4090 GPU.

As the proposed method is grounded in image steganography, the carrier is inherently an image. This design necessitates that the Multi-Gap Collaborative Fusion mechanism aligns the secret data from all modalities toward the image domain. Consequently, initializing the model with the image modality task is optimal for performance. Based on this rationale, the image modality is employed as the first modality by default in all experiments.

**Benchmarks.** To rigorously evaluate the effectiveness of the proposed method, a comprehensive comparative analysis was conducted against SOTA image steganography methods developed for different modalities. These include methods designed for text, such as SteganoGAN Zhang et al. (2019), FNNS Kishore et al. (2022), and LISO Chen et al. (2023); as well as methods tailored for image, including Baluja et al. Baluja (2017), HiDDeN Zhu (2018), Weng et al. Weng et al. (2019), UDH Zhang et al. (2020), ISN Lu et al. (2021), HiNet Jing et al. (2021), DeepMIH Guan et al. (2022), iSCMIS Li et al. (2024), and StegFormer Ke et al. (2024). To ensure a fair and objective comparison, all methods were re-trained using the same dataset employed in this study.

**Evaluation Metrics.** To assess the quality of secret/recovery pairs, we utilize Peak Signal-to-Noise Ratio(PSNR), Structural Similarity Index(SSIM) , Root Mean Square Error(RMSE), and Mean Absolute Error(MAE) as performance metrics for image and audio data. For the text data hiding scenario, the quality of stego images is evaluated employing the same metrics applied to image data, while the extraction accuracy of the embedded text is assessed using the Error Rate consistent with prior research Kishore et al. (2022); Chen et al. (2023).

## E  EFFICIENCY ANALYSIS

This section presents a systematic analysis of the efficiency of the proposed model. Build upon the Invertible Neural Network (INN), the main resource consumption of the proposed model arises from three stages: key generation, the directed customization of secret data with Multi-Gap Collaborative Fusion, and the steganography process.

In the initial steganography task, the proposed model is functionally equivalent to the original HiNet Jing et al. (2021) and the single-image DeepMIH Guan et al. (2022) model and has the identical com-

putational complexity. To accommodate new modal data, the model incorporates modal adapters. The output of each layer is then computed as

$$\mathcal{O}_{cov}(z) = Conv(z) + \beta \cdot LoRA(z),$$

where $z$ is the input of each layer. During inference, the parameters of these adapters $\theta_{ada}$ are integrated with the original convolutional layer parameters $\theta_{ori}$ to obtain new parameters, as defined by the equation

$$\theta_{new} = \theta_{ori} + \beta \cdot \theta_{ada},$$

where $\beta$ is a scaling factor that balances the contribution of the original $\theta_{ori}$ and adapter parameters $\theta_{ada}$. Thus, this integration introduces no additional computational overhead during the testing phase.

The additional computational requirements introduced by the model are mainly originate from: the Multi-Gap Collaborative Fusion mechanism for secret data customization and the key generation process. The Multi-Gap Collaborative Fusion mechanism is a three-layer INN whose parameter processing is consistent with the method described above during inference phase, resulting in a parameter overhead of only **18.75%** compared to the original HiNet. The key generation process, which is based on the ECDHE key exchange algorithm, is devoted to a two-layer MLP, contributing minimal resource consumption relative to the overall model.

A set of experiments was also conducted to evaluate computational efficiency, with the results presented in Table 4. It is observed that compared to iSCMIS Li et al. (2024), the proposed model increases FLOPs by a marginal **10.75%** yet reduces the runtime by 0.02 seconds. Compared to StegFormer StegFormer Ke et al. (2024), the proposed method uses only **23%** of the parameters and **14.73%** of the FLOPs. These results demonstrate that the proposed model achieves significant gains in multi-modal steganography performance without a considerable increase in computational resource consumption. When considered alongside its steganography performance, these findings validate the model's high effectiveness and efficiency.

## F ADDITIONAL EXPERIMENTS

**Single-image steganography.** We further evaluated the proposed model on the COCO and ImageNet datasets, and the corresponding experimental results are presented in Table 4. Consistent performance gains were observed for the proposed model on these two benchmark datasets. On the Imagenet dataset, it yields improvements of **12.18dB** and **0.7%** for cover/stego image pairs, and **12.4dB** and **0.46%** enhancement for secret/reveal pairs. Corresponding gains on the COCO dataset reached **9.1dB/0.93% (cover/stego)** and **10.19dB/0.43% (secret/reveal)**. Concurrently, significant reduction in both MAE and RMSE were demonstrated on both datasets. These findings establish the proposed model's superior steganography fidelity, demonstrating significantly enhanced quality for both stego and revealed secret images relative to benchmark methods.

**High-capacity steganography.** To assess the efficacy of the proposed method for high-capacity steganography, image data were employed as a representative case. The steganography performance of various methods was evaluated under conditions of concealing 3, 5, and 7 images. The experimental results, presented in Table 5, demonstrate the superiority of the proposed approach. Specifically, when hiding 3 images on the DIV2K dataset, the proposed method achieved gains of **8.34dB** in PSNR and **2.76%** in SSIM for cover/stego pairs, and **12.46dB** in PSNR and **3.48%** in SSIM for secret/reveal pairs. On the COCO dataset, corresponding improvements for cover/stego and secret/reveal pairs were **10.99dB/2.91%** and **10.74dB/3.68%**, respectively. Consistent performance enhancements were also observed for 5 and 7 hidden images. These results confirm the method's exceptional capability for large-capacity data hiding.

**Multimodal Secret Data steganography.** To further assess the adaptability and steganography performance of the proposed model in multimodal scenarios involving diverse data combinations, a comprehensive evaluation was conducted. The results, presented in Table 6, demonstrate the model's superior performance across all tested conditions. As a representative case, the simultaneous concealment of image and text data resulted in a **22.42dB** increase in PSNR and a **46.03%** enhancement in SSIM for cover/stego pairs. For the secret/reveal pairs, the model yielded a **26.77dB** PSNR gain and a **45.65%** SSIM gain for image data, alongside a **26.08%** reduction in error rate for text data. Significantly enhanced data hiding and extraction performance was also consistently observed under

Table 4: Numerical comparisons with different steganography methods on COCO and Imagenet datasets, highlighting the best results in **bold** and the second-best in underline.

| Method | Paras(M) | Flops(G) | Times(s) | Cover/Stego | | | | | | | |
| | | | | COCO | | | | Imagenet | | | |
| | | | | PSNR↑ | SSIM↑ | MAE↓ | RMSE↓ | PSNR↑ | SSIM↑ | MAE↓ | RMSE↓ |
|---|---|---|---|---|---|---|---|---|---|---|---|
| Baluja | 2.77 | 173.77 | 0.24 | 36.38 | 0.9563 | 5.98 | 7.43 | 36.59 | 0.9520 | 5.61 | 5.41 |
| UDH | 17.40 | 50.62 | 0.22 | 38.90 | 0.9650 | 2.77 | 2.90 | 38.96 | 0.9624 | 2.75 | 2.88 |
| ISN | 2.99 | 196.46 | 0.46 | 37.95 | 0.9751 | 2.76 | 3.23 | 40.13 | 0.9748 | 1.95 | 2.51 |
| HiNet | 4.05 | 20.59 | 0.36 | 39.01 | 0.9844 | 2.09 | 2.96 | 44.61 | 0.9927 | 1.52 | 1.63 |
| DeepMIH | 5.40 | 22.13 | 0.30 | 40.30 | 0.9805 | 2.83 | 4.14 | 40.31 | 0.9800 | 2.87 | 4.16 |
| iSCMIS | 5.48 | 27.63 | 0.34 | 41.53 | 0.9818 | 2.53 | 3.78 | 40.31 | 0.9818 | 2.59 | 3.79 |
| StegFormer | 34.96 | 207.78 | 0.29 | 42.62 | 0.9897 | 2.09 | 2.94 | 42.87 | 0.9875 | 1.92 | 2.83 |
| Ours | 8.04 | 30.60 | 0.32 | **51.72(9.1↑)** | **0.9990(0.0093↑)** | **0.49(1.6↓)** | **0.69(2.25↓)** | **56.79(12.18↑)** | **0.9997(0.0070↑)** | **0.26(1.26↓)** | **0.38(1.25↓)** |

| Method | Srcret/Reveal | | | | | | | |
| | COCO | | | | Imagenet | | | |
| | PSNR↑ | SSIM↑ | MAE↓ | RMSE↓ | PSNR↑ | SSIM↑ | MAE↓ | RMSE↓ |
|---|---|---|---|---|---|---|---|---|
| Baluja | 35.01 | 0.9341 | 6.52 | 8.00 | 34.13 | 0.9247 | 5.31 | 8.37 |
| UDH | 35.07 | 0.8220 | 3.77 | 4.67 | 35.39 | 0.8252 | 3.73 | 4.58 |
| ISN | 36.58 | 0.9016 | 3.04 | 3.78 | 37.73 | 0.9548 | 2.97 | 3.31 |
| HiNet | 44.05 | 0.9952 | 1.17 | 1.70 | 46.78 | 0.9952 | 1.94 | 2.74 |
| DeepMIH | 36.55 | 0.9613 | 5.09 | 6.48 | 36.63 | 0.9604 | 4.16 | 6.07 |
| iSCMIS | 39.47 | 0.9754 | 3.74 | 5.48 | 39.44 | 0.9718 | 3.79 | 5.48 |
| StegFormer | 42.04 | 0.9884 | 2.74 | 4.11 | 42.39 | 0.9862 | 2.24 | 3.47 |
| Ours | **54.24(10.19↑)** | **0.9995(0.0043↑)** | **0.36(0.81↓)** | **0.55(0.1.15↓)** | **59.18(12.4↑)** | **0.9998(0.0046↑)** | **0.22(1.72↓)** | **0.28(1.86↓)** |

Table 5: Numerical comparisons of diverse steganography approaches on the DIV2K and COCO datasets for multi-image hiding, highlighting the best results in **bold** and the second-best in underline.

| N | Method | DIV2K | | | | COCO | | | |
| | | Cover/Stego | | Secret/Reveal | | Cover/Stego | | Secret/Reveal | |
| | | PSNR↑ | SSIM↑ | PSNR↑ | SSIM↑ | PSNR↑ | SSIM↑ | PSNR↑ | SSIM↑ |
|---|---|---|---|---|---|---|---|---|---|
| 3 | ISN | 31.23 | 0.8203 | 30.05 | 0.8401 | 33.53 | 0.8262 | 31.85 | 0.8110 |
| | DeepMIH | 31.29 | 0.7685 | 27.30 | 0.8321 | 33.99 | 0.8192 | 29.56 | 0.8373 |
| | iSCMIS | 33.39 | 0.7994 | 30.78 | 0.8302 | 34.75 | 0.8636 | 33.82 | 0.8046 |
| | StegFormer | 39.67 | 0.9709 | 36.81 | 0.9642 | 38.25 | 0.9691 | 39.21 | 0.9621 |
| | Ours | **48.01(8.34↑)** | **0.9985(0.0276↑)** | **49.27(12.46↑)** | **0.9990(0.0348↑)** | **49.24(10.99↑)** | **0.9982(0.0291↑)** | **49.95(10.74↑)** | **0.9989(0.0368↑)** |
| 5 | ISN | 26.74 | 0.6923 | 27.37 | 0.7131 | 30.52 | 0.7772 | 29.08 | 0.7903 |
| | DeepMIH | 29.48 | 0.0.6925 | 22.19 | 0.7164 | 32.51 | 0.7926 | 26.31 | 0.7862 |
| | iSCMIS | 30.68 | 0.7124 | 22.07 | 0.7142 | 32.68 | 0.8125 | 25.99 | 0.7851 |
| | StegFormer | 35.12 | 0.9317 | 33.88 | 0.9290 | 34.97 | 0.9432 | 33.65 | 0.9168 |
| | Ours | **43.08(7.96↑)** | **0.9968(0.0651↑)** | **43.39(9.51↑)** | **0.9973(0.0683↑)** | **44.78(9.81↑)** | **0.9961(0.0529↑)** | **45.04(11.39↑)** | **0.9957(0.0798↑)** |
| 7 | ISN | 24.28 | 0.6759 | 24.91 | 0.6895 | 28.48 | 0.7567 | 27.80 | 0.7830 |
| | DeepMIH | 27.32 | 0.6813 | 20.94 | 0.6083 | 30.21 | 0.7751 | 24.54 | 0.7597 |
| | iSCMIS | 28.46 | 0.6942 | 21.26 | 0.6054 | 30.35 | 0.7838 | 26.13 | 0.7905 |
| | StegFormer | 35.05 | 0.9224 | 32.61 | 0.9119 | 33.95 | 0.9175 | 32.28 | 0.9017 |
| | Ours | **41.79(6.74↑)** | **0.9961(0.0737↑)** | **42.06(9.45↑)** | **0.9963(0.0844↑)** | **42.97(9.02↑)** | **0.9947(0.0772↑)** | **42.27(9.99↑)** | **0.9952(0.0935↑)** |

the other three multimodal combinations. These findings substantiate the model's robust steganography capabilities and its strong adaptability to complex multimodal environments.

## G MORE ANALYSIS

**Steganographic analysis.** The anti-steganalysis ability is a critical metric for assessing the security of image steganography, as it measures the likelihood that stego images can be distinguished from reference images using steganalysis tools. To evaluate the anti-steganalysis capability of our model alongside other methods, we employ the open-source steganalysis tool StegExpose Boehm (2014) and three steganalysis networks: SRNet Boroumand et al. (2018), XuNet Xu et al. (2016), and YeNet Ye et al. (2017). Lower detection accuracy and a smaller area under curve (AUC) indicates better security performance. The evaluate results are presented in Figure 7 and Table 2 respectively. These steganalysis results indicate that the proposed model achieves superior anti-steganalysis performance compared to other SOTA methods.

**The Security of Concealed Secret Data.** The Multi-Gap Collaborative Fusion mechanism seamlessly integrates the symmetric key $k_{sym}$ to simultaneously customize and encrypt multimodal secret data during mitigating the information conflict. Its security enhancement efficacy was also

Table 6: Numerical comparisons of diverse steganography approaches in multimodal environments, highlighting the best results in **bold** and the second-best in underline.

| Method | Cover/Stego | | Secret/Reveal | | |
|---|---|---|---|---|---|
| | **Image+Text** | | Image | | Text |
| | PSNR↑ | SSIM↑ | PSNR↑ | SSIM↑ | Error Rate(%)↓ |
| ISN | 16.57 | 0.4821 | 17.04 | 0.4853 | 27.53 |
| HiNet | 17.08 | 0.4872 | 17.52 | 0.4905 | 28.06 |
| DeepMIH | 17.52 | 0.5019 | 18.60 | 0.5113 | 27.91 |
| iSCMIS | 18.01 | 0.5167 | 18.14 | 0.5198 | 25.82 |
| StegFormer | 18.27 | 0.5204 | 19.01 | 0.5179 | 26.16 |
| Ours | **40.69(22.42↑)** | **0.9807(0.4603↑)** | **45.78(26.77↑)** | **0.9763(0.4565↑)** | **0.08(26.08↓)** |

| Method | Cover/Stego | | Secret/Reveal | | | |
|---|---|---|---|---|---|---|
| | **Image+Audio** | | Image | | Audio | |
| | PSNR↑ | SSIM↑ | PSNR↑ | SSIM↑ | PSNR↑ | SSIM↑ |
| ISN | 33.68 | 0.8582 | 32.15 | 0.8437 | 28.58 | 0.8013 |
| HiNet | 35.74 | 0.8791 | 34.59 | 0.8681 | 29.86 | 0.8357 |
| DeepMIH | 36.68 | 0.8773 | 35.87 | 0.8724 | 30.29 | 0.8329 |
| iSCMIS | 37.92 | 0.9091 | 37.14 | 0.9083 | 30.05 | 0.8502 |
| StegFormer | 38.99 | 0.9447 | 39.43 | 0.9379 | 31.85 | 0.8429 |
| Ours | **48.51(9.52↑)** | **0.9937(0.0490↑)** | **48.79(9.36↑)** | **0.9914(0.0535↑)** | **43.92(12.07↑)** | **0.9852(0.1350↑)** |

| Method | Cover/Stego | | Secret/Reveal | | |
|---|---|---|---|---|---|
| | **Text+Audio** | | Audio | | Text |
| | PSNR↑ | SSIM↑ | PSNR↑ | SSIM↑ | Error Rate(%)↓ |
| ISN | 15.72 | 0.4711 | 16.44 | 0.4707 | 28.79 |
| HiNet | 17.01 | 0.4708 | 16.82 | 0.4749 | 28.54 |
| DeepMIH | 17.47 | 0.4873 | 17.41 | 0.4826 | 27.99 |
| iSCMIS | 17.79 | 0.4937 | 17.98 | 0.4951 | 26.57 |
| StegFormer | 18.10 | 0.5109 | 18.43 | 0.5317 | 25.94 |
| Ours | **38.97(20.87↑)** | **0.9826(0.4717↑)** | **39.57(21.14↑)** | **0.9683(0.4366↑)** | **0.13(25.81↓)** |

| Method | Cover/Stego | | Secret/Reveal | | | | |
|---|---|---|---|---|---|---|---|
| | **Image+Text+Audio** | | Image | | Text | Audio | |
| | PSNR↑ | SSIM↑ | PSNR↑ | SSIM↑ | Error Rate(%)↓ | PSNR↑ | SSIM↑ |
| ISN | 14.57 | 0.4130 | 14.25 | 0.4029 | 34.57 | 13.88 | 0.4001 |
| HiNet | 15.29 | 0.4207 | 14.83 | 0.4187 | 32.81 | 14.07 | 0.4073 |
| DeepMIH | 15.77 | 0.4214 | 15.49 | 0.4195 | 30.49 | 14.72 | 0.4115 |
| iSCMIS | 16.01 | 0.4305 | 16.47 | 0.4383 | 29.14 | 15.52 | 0.4293 |
| StegFormer | 16.23 | 0.4293 | 16.50 | 0.4311 | 28.98 | 15.76 | 0.4300 |
| Ours | **36.09(19.86↑)** | **0.9653(0.5348↑)** | **42.87(26.37↑)** | **0.9537(0.5154↑)** | **0.79(28.19↓)** | **38.74(22.98↑)** | **0.9277(0.4977↑)** |

empirially validated, as illustrated in Table 7 and Figure 9. These results confirm that high-fidelity reconstruction was achieved only with the authenticated key, whereas severe errors occurred when an incorrect key was used. These findings underscore the essential role of the proposed Multi-Gap Collaborative Fusion mechanism in enhancing the security of the concealed multimodal secret data.

**Effect of the Encryption Loss $\mathcal{L}_e$.** The $\mathcal{L}_e$ is introduced to mitigate reconstruction errors between decrypted secret data $x_{rev}$ and the original secret data $x_{sec}$, where auxiliary variable $z_{aux}$ introduces perturbations during the decryption process. As evidenced by Table 3, the proposed $\mathcal{L}_e$ substantially enhances reconstruction fidelity, yielding PSNR improvements of **5.56dB** for cover/stego pairs and **4.69dB** for secret/revealed pairs, with corresponding SSIM gains of **0.18%** and **0.21%** respectively. Concurrently, MAE and RMSE exhibit marked reductions of **0.53/0.76** and **0.35/0.48** for cover/stego and secret/revealed pairs respectively. These quantitative improvements substantiate the efficacy of the proposed Encryption Loss $\mathcal{L}_e$ in enhancing steganography performance.

Table 7: Comparative Performance of Secret Image Extraction with Correct and Incorrect Keys.

|  | Cover/Stego | Correct Key | Random Key | Public Key |
|---|---|---|---|---|
| PSNR↑ | 50.72 | 53.10 | 8.71 | 7.67 |
| SSIM↑ | 0.9987 | 0.9996 | 0.1810 | 0.0856 |
| MAE↓ | 0.55 | 0.41 | 86.99 | 97.2 |
| RMSE↓ | 0.78 | 0.61 | 98.73 | 110.84 |

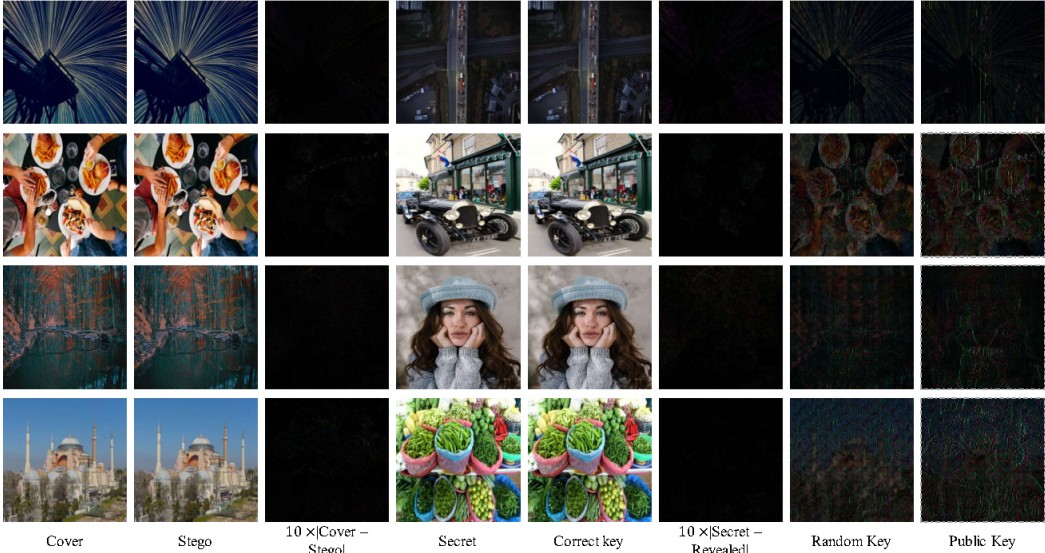

Figure 9: Visual results on DIV2K dataset. The secret images are revealed with three types of keys: correct keys, random keys and the public keys tied to secret images.

## H    ADDITIONAL QUALITATIVE RESULTS

The qualitative comparison outcomes for the stego and recovery images of our model and other models are presented in Figure 10, Figure 11, Figure 12, and Figure 13. Figure 10, Figure 11, Figure 12, and Figure 13 present visual comparisons for image-in-image, text-in-image, audio-in-image, and multimodal secret data (image+text+audio) concealing tasks, respectively. Each figure includes the residual maps (magnified by a factor of 10 for clarity) between the cover and stego images, as well as the original and revealed secret data. The comparisons demonstrate the superior performance of the proposed method across all tasks, outperforming existing approaches in both single-modal and multimodal secret data concealing scenarios. These results confirm the model's efficacy and strong adaptability in multimodal secret data steganography.

## I    USE OF LARGE LANGUAGE MODELS (LLMS)

This manuscript underwent language polishing and editorial refinement with the assistance of a large language model (LLM). The model's function was solely to enhance the expressive quality of the author's original writing without contributing to any core research components such as ideation, experimental design, data analysis, or technical interpretation.

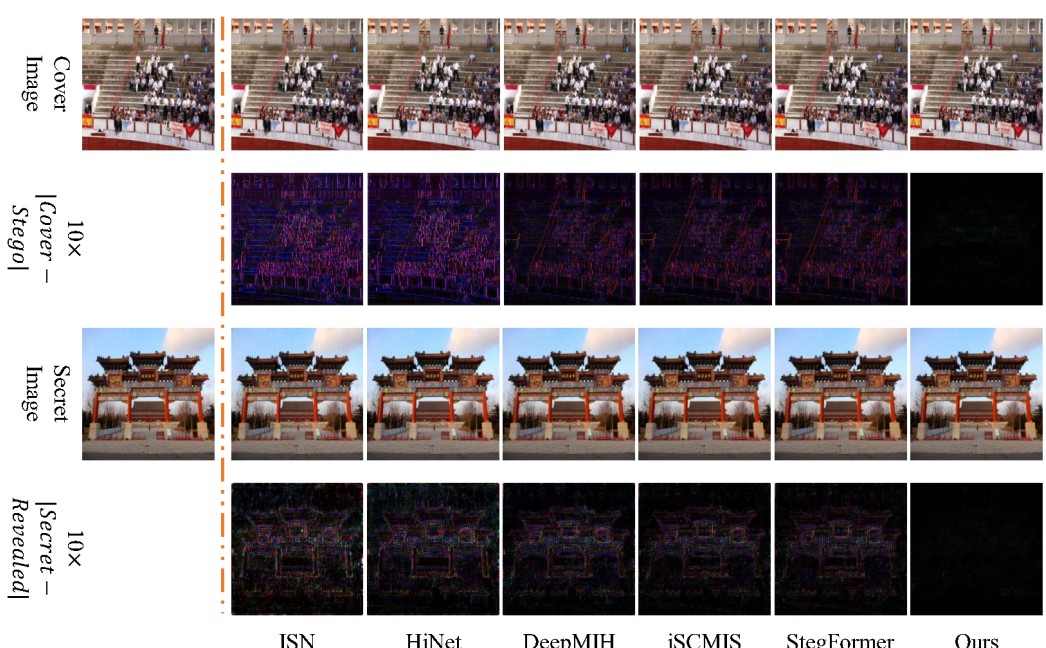

Figure 10: Visual comparisons of our model with other steganography models for concealing image on the DIV2K datasets.

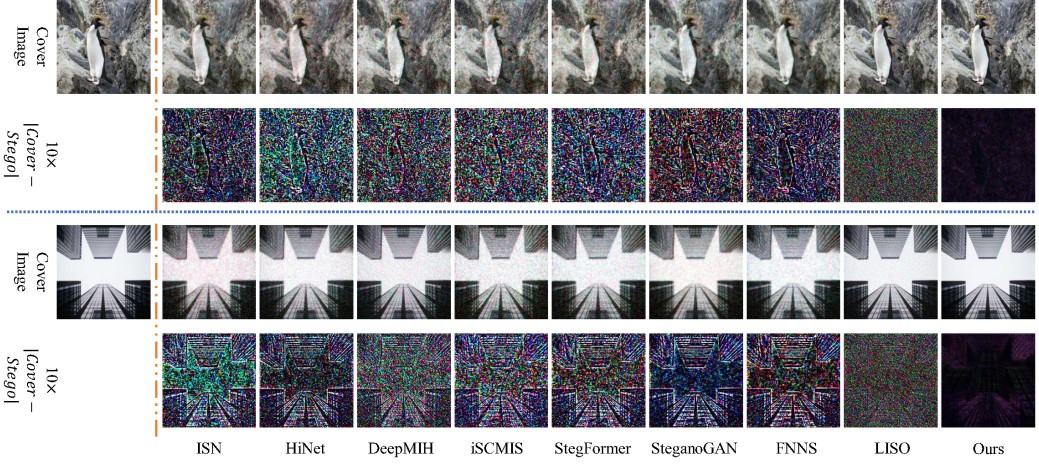

Figure 11: Visual comparisons of our model with other steganography models for concealing randomly generated binary text data.

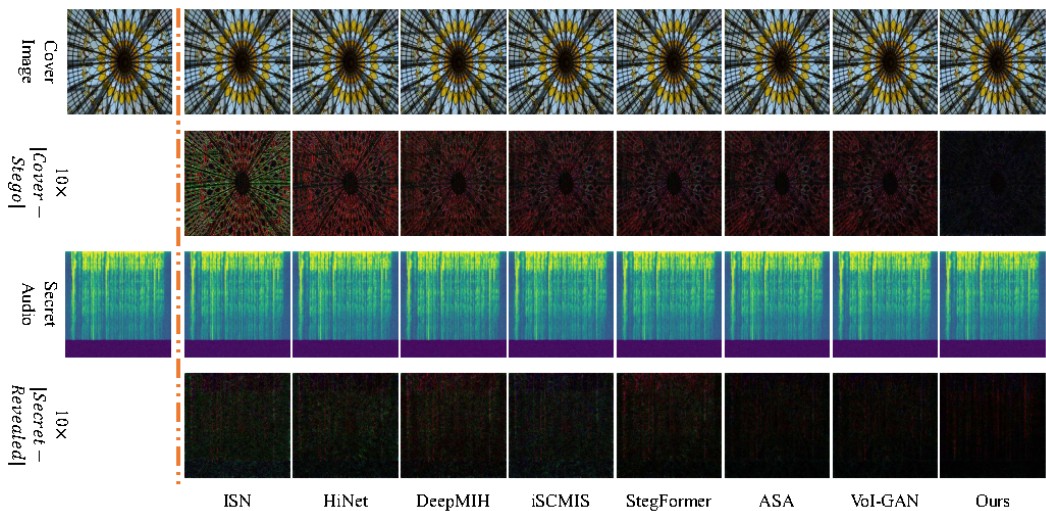

Figure 12: Visual comparisons of our model with other steganography models for concealing audio data on the Dani-Voice datasets.

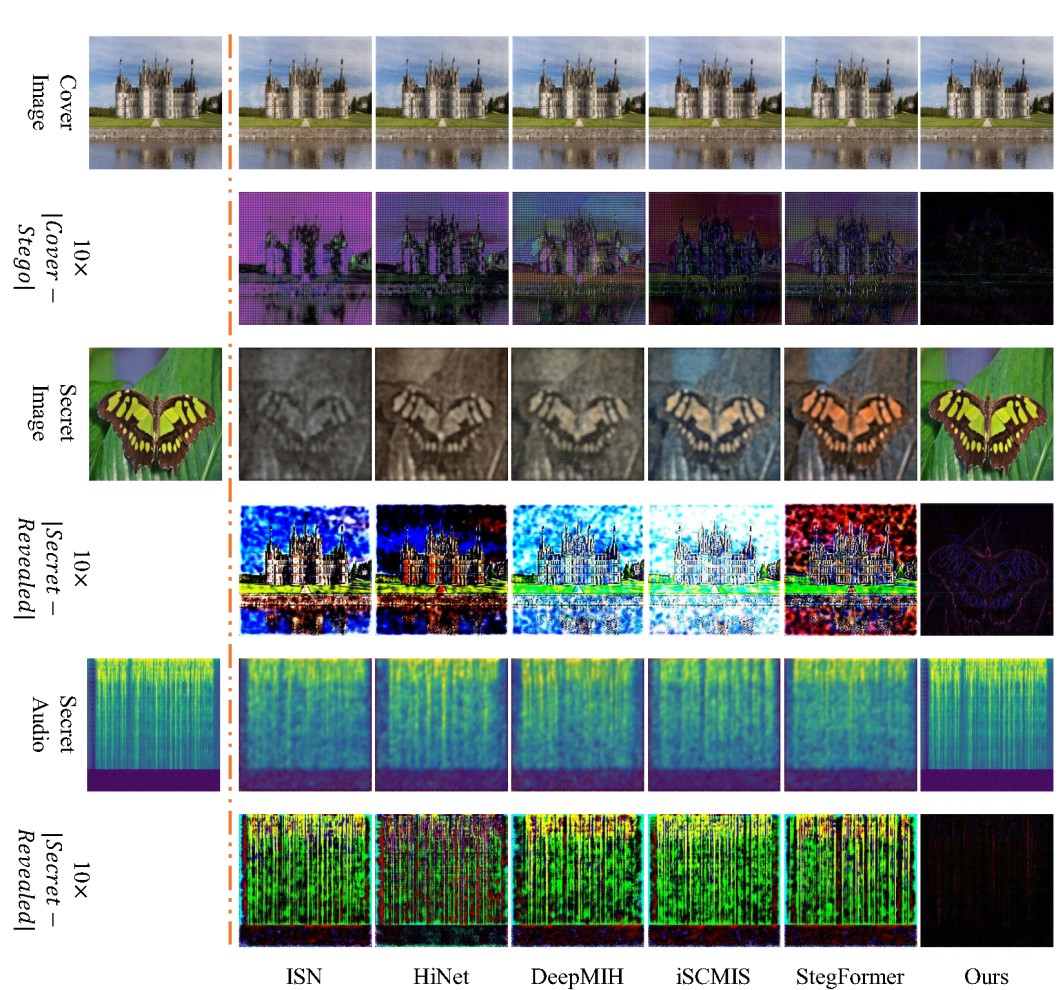

Figure 13: Visual comparisons of our model with other steganography models for concealing three modalities of secret data on the DIV2K and Dani-Voice datasets. The text data is randomly generated binary data.

