# OpenReview forum: "A Unified Framework for Multimodal Secret Data Steganography"
_ICLR.cc/2026/Conference — ICLR 2026 Conference Withdrawn Submission_

### Official Review · Reviewer_oYYX · 2025-10-16

**Soundness:** 3
**Presentation:** 2
**Contribution:** 2
**Rating:** 4
**Confidence:** 4

**Summary:**

The paper proposes a unified steganography model that can conceal multimodal data such as images, text, and audio into a cover image. The authors use a LoRA adapter to mitigate gradient conflicts between different modalities and employ continual learning to alleviate the problem of forgetting multimodal knowledge during training.

**Strengths:**

The paper presents a steganography framework based on continual learning and LoRA, which enables hiding multimodal data (e.g., image, text, audio) into cover images.

**Weaknesses:**

The advantage of using a single model to hide multimodal data into cover images is not clearly demonstrated. If the benefit lies in computational efficiency, the paper should provide quantitative comparisons, including the model parameters, training time under the same computational resources, inference FLOPs, and GPU memory requirements. If the benefit lies in steganographic performance, although the proposed method achieves the best results in Table 1, the data are not reliable (see the next weakness for details).

I am quite certain that both the proposed method and the compared methods have not quantized the generated stego images. Specifically, the conceal network outputs color stego images where each pixel is represented by 3 × 32-bit floating-point values, without being quantized to 3 × 8-bit before being input to the reveal network. This makes the results unrealistic for practical use. Quantized results should be provided to demonstrate that multimodal fusion indeed enhances steganographic performance.

Quantized results without continual learning should also be presented. I believe that, in the proposed framework, the LoRA adapter plays a more critical role than continual learning in resolving multimodal conflicts.

Some figures are too small to observe details clearly.

“Concealing network” and “revealing network” are more grammatically correct than “conceal network” and “reveal network.”

**Questions:**

In the Encryption Loss, it is unclear why the distance between the frequency representation of the cover image and the auxiliary variable should be minimized. Please clarify the rationale for this design.

The use of public and private keys to generate a symmetric key is confusing. Please explain the purpose of this process and how it contributes to the system.

---

### Official Review · Reviewer_Htc4 · 2025-10-25

**Soundness:** 3
**Presentation:** 2
**Contribution:** 2
**Rating:** 4
**Confidence:** 4

**Summary:**

This paper proposes a unified model capable of concealing and recovering multimodal secret data (text, image, audio) within cover images. The framework integrates continual learning to mitigate catastrophic forgetting across modalities and introduces a Multi-Gap Collaborative Fusion mechanism to reduce intra-/inter-modal conflicts by using the cover image as an anchor for multimodal fusion.

**Strengths:**

1.	Novel integration of steganography and continual learning concepts.
2.	Clear articulation of challenges (forgetting and modal conflict).
3.	Security-oriented approach via directional fusion and encryption key use.

**Weaknesses:**

1. Lack of quantitative rigor: Most claims (reversibility, imperceptibility, robustness) are qualitative; no PSNR, SSIM, or BER metrics are provided.
2. Scalability concerns: Continual learning and multi-gap fusion may scale poorly as modality count increases. Computational overhead is not discussed.
3. Incomplete methodological detail: It is unclear how the continual learning component is implemented—whether via rehearsal, EWC, or knowledge distillation.
4. Security analysis missing: Although encryption is mentioned, no formal cryptographic evaluation (entropy or key-space size) is presented.
5. Overstated claims: The authors assert “perfect imperceptibility,” which is unrealistic without quantitative support.
6. Presentation clarity: The model architecture diagrams are abstract; notation and variable definitions are not consistent.

**Questions:**

1. What quantitative metrics demonstrate superiority—PSNR, BER, or imperceptibility measures?
2. How does the model balance embedding capacity and image quality?
3. Is the continual learning module based on replay or regularization?
4. Can the approach handle four or more modalities efficiently?
5. How is the encryption implemented—before or after embedding?

---

### Official Review · Reviewer_9RmH · 2025-10-28

**Soundness:** 2
**Presentation:** 3
**Contribution:** 2
**Rating:** 4
**Confidence:** 2

**Summary:**

This paper introduces a unified steganography framework designed to conceal diverse data modalities (image, text, audio) within a single cover image. The core challenge in this domain is twofold(catastrophic forgetting, inter-modal information conflicts). The proposed method tackles these issues by reframing the problem through a continual learning paradigm to preserve knowledge, and by introducing a novel fusion mechanism that uses the cover image as an anchor to mitigate data conflicts. Experiments demonstrate that this approach achieves state-of-the-art performance in both fidelity and security for multimodal steganography.

**Strengths:**

1. The paper is well-written and easy to follow, providing clear and detailed information on the experimental settings.

2. The core idea of the MGCF mechanism, which uses the cover image as an anchor to customize secret data, is a good contribution.

3. The method's performance is well-demonstrated through a comprehensive experimental design.

**Weaknesses:**

1. The justification for using continual learning as the core premise is weak. Continual learning typically assumes a dynamic environment where new tasks are continuously added or datasets constantly grow. However, the multimodal steganography problem described here (image, text, audio) involves a very small, fixed number of tasks. In this scenario, catastrophic forgetting would not even occur if all modal datasets were simply pooled and trained at once using multi-task learning. Therefore, it is questionable whether CL is a necessary approach for this problem.

2. The proposed framework is a complex combination of two major techniques, continual learning with LoRA adapters and MGCF. However, there is a lack of ablation studies to clearly isolate the contribution of each component to the final performance. This makes it difficult to assess the true effect of each mechanism.

3. The baseline methods in Tables 1 and 6 are inherently designed for single-modality hiding. Directly comparing these to the proposed multimodal-capable model is inappropriate. A fairer comparison would require baselines such as a simple multi-task model trained on all modalities at once, or the original single-modality models with the same continual learning technique applied.

4. Minor points

- The paper omits a discussion of relevant prior work. Several recent studies [1, 2] have addressed multimodal steganography using Implicit Neural Representations (INR), yet this paper includes no mention or comparison of these approaches.

[1] Han, G., Lee, D. J., Hur, J., Choi, J., & Kim, J. (2023, October). Deep cross-modal steganography using neural representations. In 2023 IEEE International Conference on Image Processing (ICIP) (pp. 1205-1209). IEEE.

[2] Song, S., Yang, S., Yoo, C. D., & Kim, J. (2024, September). Implicit Steganography Beyond the Constraints of Modality. In European Conference on Computer Vision (pp. 289-304). Cham: Springer Nature Switzerland.

- The paper suffers from readability issues. Figures 1, 2, and 3, in particular, are too small and have low resolution, making them very difficult to understand. The readability of visual aids is critical for effectively communicating a paper's content.

**Questions:**

1. Given that the number of modalities is fixed, what is the specific reason for using continual learning instead of multi-task learning, which would avoid catastrophic forgetting from the start?

2. Could you provide ablation studies that isolate the performance contributions of the MGCF and the continual learning?

3. Does the order of continual learning affect the final performance?

---

### Official Review · Reviewer_zJXK · 2025-10-31

**Soundness:** 1
**Presentation:** 2
**Contribution:** 2
**Rating:** 2
**Confidence:** 4

**Summary:**

This paper addresses a multimodal steganography technique for hiding and recovering various types of secret data within images. Previous studies mainly used models specialized for a single modality or required separate sequential training for different modal data. In contrast, this work proposes an integrated multimodal steganography model that can handle multiple modalities—such as images, audio, and text—within a unified framework. To mitigate the catastrophic forgetting problem that may occur in this process, the authors adopt the concept of continual learning and employ modality-specific LoRA-based adapters to preserve performance on previously learned modalities. In addition, the authors design a Multi-Gap Collaborative Fusion (MGCF) mechanism to reduce distributional discrepancies between cover images and secret data, and introduce a key-conditioned invertible mapping to achieve more stable and secure hiding. Experimental results show that the proposed method outperforms existing steganography techniques in terms of secret data reconstruction quality, minimal cover image distortion, and steganalysis resistance, achieving state-of-the-art results.

**Strengths:**

While most prior image-based steganography research has focused on single-modality or limited-modality combinations, this paper interprets the problem of hiding multimodal secret data as a task sequence scenario in which different modalities are given sequentially, and attempts to solve it from a continual learning perspective. This idea is novel. In particular, using modality-specific LoRA adapters to extend functionality without retraining the entire network for each modality is an efficient and practical design choice. Furthermore, the proposed method demonstrates meaningful performance gains over existing steganography techniques in terms of both hiding quality and security.

**Weaknesses:**

The paper lacks clarity in concept definitions and terminology, and the experimental setup is insufficiently described. Figures 2 and 3 do not provide adequate details about the datasets and experimental settings used. Likewise, in the main experimental and ablation tables, it is unclear which modality’s cover data hides which modality’s secret data, and how the experimental protocols are organized. The term “secret optimize module” in Table 3 does not match the terminology in the Method section and needs to be unified. Several terms—such as “intra- and inter-modal information conflict,” “structured linkage,” “multi-gap collaborative fusion,” and “dynamic multimodal setting”—are not standard and lack precise definitions. Clear conceptual correspondence to established academic terminology is needed.

The paper’s core claims are not consistent with its implementation and experiments:
(a) The paper claims that a single unified model handles multiple modalities, but in practice, separate LoRA adapters are trained per modality. This constitutes a multi-adapter system with modality-specific parameter sets, which cannot be regarded as a truly unified model.
(b) The main goal of continual learning is to progressively learn new information while retaining existing knowledge, but the proposed approach simply expands parameters by adding LoRA layers for each modality. This is not a standard continual learning methodology and poses scalability limitations.
(c) Although the paper claims to address catastrophic forgetting from a continual learning perspective, there are no experiments to verify this—such as forgetting metrics or retention curves.
(d) The paper also claims that MGCF and modal adapters enable cross-modal interaction, but no experimental evidence supports this claim.

**Questions:**

In Table 1, the evaluation metrics for Text cover/stego are PSNR, SSIM, MAE, and RMSE—image-based reconstruction metrics—rather than Error Rate (%). Why are image metrics used for text data?

Section 3.3 claims that Multi-Gap Collaborative Fusion aligns secret features toward cover features and expresses this as a distance constraint. However, the loss function section does not include any distance-based loss term that induces alignment. What specific distance function is used, and how is alignment learned without an explicit loss term?

---

### Note · Authors · 2025-12-14

I have read and agree with the venue's withdrawal policy on behalf of myself and my co-authors.